# Video-based interventions to improve self-assessment accuracy among physicians: A systematic review

Chandni Pattni[1]ᵒ, Michael Scaffidi[1,2]ᵒ, Juana Li[1], Shai Genis[1], Nikko Gimpaya[1], Rishad Khan[1], Rishi Bansal[1], Nazi Torabi[3], Catharine M. Walsh[4,5,6,7]ᵒ, Samir C. Grover[1,8]*

1 Division of Gastroenterology, St. Michael's Hospital, University of Toronto, Toronto, Canada, 2 Department of Medicine, Queen's University, Kingston, Canada, 3 Gerstein Science Information Centre, University of Toronto, Toronto, Canada, 4 Department of Paediatrics, Hospital for Sick Children, University of Toronto, Toronto, Canada, 5 Department of Medicine, Hospital for Sick Children, University of Toronto, Toronto, Canada, 6 The Wilson Centre, University of Toronto, Toronto, Canada, 7 Division of Gastroenterology, Hepatology and Nutrition, Hospital for Sick Children, University of Toronto, Toronto, Canada, 8 Li Ka Shing Knowledge Institute, Toronto, Canada

ᵒ These authors contributed equally to this work.
* samir.grover@utoronto.ca

## Abstract

### Purpose

Self-assessment of a physician's performance in both procedure and non-procedural activities can be used to identify their deficiencies to allow for appropriate corrective measures. Physicians are inaccurate in their self-assessments, which may compromise opportunities for self-development. To improve this accuracy, video-based interventions of physicians watching their own performance, an experts' performance or both, have been proposed to inform their self-assessment. We conducted a systematic review of the effectiveness of video-based interventions targeting improved self-assessment accuracy among physicians.

### Materials and methods

The authors performed a systematic search of MEDLINE, Embase, EBM reviews, and Scopus databases from inception to August 23, 2022, using combinations of terms for "self-assessment", "video-recording", and "physician". Eligible studies were empirical investigations assessing the effect of video-based interventions on physicians' self-assessment accuracy with a comparison of self-assessment accuracy pre- and post- video intervention. We defined self-assessment accuracy as a "direct comparison between an external evaluator and self-assessment that was quantified using formal statistical analysis". Two reviewers independently screened records, extracted data, assessed risk of bias, and evaluated quality of evidence. A narrative synthesis was conducted, as variable outcomes precluded a meta-analysis.

**Data Availability Statement:** All relevant data are within the manuscript and its Supporting information files.

**Funding:** Funding/Support: CMW holds an Early Researcher Award from the Ontario Ministry of Research and Innovation. The funders had no role in the design and conduct of the review, decision to publish and preparation, review, or approval of the manuscript.

**Competing interests:** Other disclosures: Rishad Khan has received research grants from AbbVie and Ferring Pharmaceuticals and research funding from Pendopharm. Samir C. Grover has received research grants and personal fees from AbbVie and Ferring Pharmaceuticals, personal fees from Takeda, education grants from Janssen, and has equity in Volo Healthcare. This does not alter our adherence to PLOS ONE policies on sharing data and materials.

## Results

A total of 2,376 papers were initially retrieved. Of these, 22 papers were selected for full-text review; a final 9 studies met inclusion criteria for data extraction. Across studies, 240 participants from 5 specialties were represented. Video-based interventions included self-video review (8/9), benchmark video review (3/9), and/or a combination of both types (1/9). Five out of nine studies reported that participants had inaccurate self-assessment at baseline. After the intervention, 5 of 9 studies found a statistically significant improvement in self-assessment accuracy.

## Conclusions

Overall, current data suggests video-based interventions can improve self-assessment accuracy. Benchmark video review may enable physicians to improve self-assessment accuracy, especially for those with limited experience performing a particular clinical skill. In contrast, self-video review may be able to provide improvement in self-assessment accuracy for more experience physicians. Future research should use standardized methods of comparison for self-assessment accuracy, such as the Bland-Altman analysis, to facilitate meta-analytic summation.

## Introduction

Self-assessment, the evaluation of one's own performance compared to a standard, can be used to identify specific deficiencies in a physician's performance of relevant procedural and non-procedural skills. Ultimately, this would allow for appropriate corrective measures [1–3]. The current literature, however, suggests that physicians are inaccurate in their self-assessments, which may compromise opportunities for self-development [4,5]. In particular, self-assessment accuracy, the extent to which one's own evaluation corresponds to an external standard, is inaccurate among medical subspecialty physicians and surgeons alike [5–8]. Inaccurate self-assessment may impede opportunities for professional growth, allowing for mistaken overconfidence in one's own performance [9,10].

To mitigate the risks from inaccurate self-assessment and optimize opportunities for self-development, video-based interventions have been proposed. Video-based interventions involve physicians watching their own performance, an experts' performance or both, in order to inform their self-assessment [11–13]. It has been proposed that these interventions work by facilitating feedback to the individual via visual cues, which allows for more informed assessment [12]. To date, however, a comprehensive evaluation of these interventions and their outcomes has not been conducted, as there is no systematic review on the topic [14].

We conducted a systematic review of the effectiveness of video-based interventions targeting improved self-assessment accuracy among all physicians, in order to inform educational and clinical practices as well as provide suggestions for future research.

## Materials and methods

We prospectively registered this systematic review in the PROSPERO International Prospective Register of Systematic Reviews (CRD42020161954) and followed reporting guidelines outlined in the Preferred Reporting Items for Systematic Reviews and Meta-Analyses (PRISMA) statement [15].

## Definitions

We defined self-assessment accuracy as a direct comparison between an external evaluator and self-assessment that was quantified using formal statistical analysis. Specifically, we considered "accurate self-assessment" according to how it was defined in the included papers.

## Data sources and searches

Using a search strategy developed by an academic health sciences librarian (N.T.), we systematically searched four bibliographic databases—Ovid MEDLINE (including Epub ahead of print, in-process, and other unindexed citations), Ovid Embase, EBM Reviews, and Scopus—from database inception through August 23, 2022. We conducted an initial database search on September 17, 2019, and updated it on August 11, 2020, August 20, 2021 and August 23, 2022. Search terms included the concepts of "self-assessment", "video recording" and "physicians", while also using a combination of subject headings and key words (see S1 File). Additionally, we searched the grey literature through August 2022 using the following: metaRegister of controlled trials (active and archived registers); the WHO ICTRP portal (http://apps.who.int/trialsearch/); Clinical Trials.gov (https://clinicaltrials.gov/); and the Cochrane review database. Finally, we searched abstracts and proceedings of major medical education meetings including AMEE: The Association for Medical Education in Europe Conference (2015–22), the International Conference on Residency Education (2014–22); and the Canadian Conference on Medical Education (2018–22).

## Eligibility criteria

We included all randomized controlled trials (RCTs) and non-randomized before-after studies investigating the effect of video-based interventions to improve self assessment accuracy among physicians regardless of career stage (i.e. post-graduate trainees/residents/fellows and/or practicing physicians). We only examined studies in which the majority of participants were physicians, as we aimed to represent our findings among only fully licensed, independent practitioners. (Note: we only included studies that had medical students if the total proportion of medical students was <50% among all participants and the remainder of the participants were attending physicians and/or post-graduate trainees). We excluded studies if they only examined external or self-assessment without comparison to one another, assessed non-procedural tasks (e.g. communication skills), did not use a video-based intervention, was not a primary study (e.g. review), only involved medical students and/or non-physicians, published only in abstract form, or if they were published in any language other than English. We made no exclusions based on the date of publication.

## Study selection

All retrieved studies were screened in a 2-stage process, beginning with titles and abstracts, for which inter-rater agreement was 92.3% ($\kappa = 0.36$), then proceeding to full text before determining eligibility. At each level, the studies were screened independently by two authors (CP, MAS) using Colandr, an open-source, research and evidence synthesis tool [16,17]. Chance-corrected agreement on inclusion after full-text screening between raters was high at 95.4% ($\kappa = 0.91$). Any discrepancies were resolved by consensus discussion and adjudication in the presence of a third reviewer (NG).

## Data extraction and synthesis

Two authors (CP, MAS) independently extracted the information from all eligible studies using a standardized data extraction sheet with Excel 2016 (Microsoft Corporation, Redmond, Washington). We extracted data on the following:

- Study demographics, including title, authors, publication date, journal of publication, study design (observation or interventional), task (procedural or non-procedural and exact task), study setting (clinical or simulated);

- Participant and external observer details including, number of participants, experience level, subspecialty, metric of inter-rater agreement among external observers (e.g. Cohen's kappa)

- Type of video interventions and assessment, including self-video and/or benchmark/expert video), self/external assessment tools and validity evidence of the tools, timeframe of intervention with respect to post-intervention self-assessment

- Type of outcomes, including statistical method for comparison of self-assessment accuracy, self-assessment accuracy at baseline with associated study metric, and impact of video-based intervention on self-assessment accuracy with p-values where provided.

Any disagreements were resolved by consensus discussion. If a study was missing data, we contacted the authors of that study in an effort to obtain the missing data.

## Data analysis

The primary outcome was the impact of the video-based intervention on self-assessment accuracy. Due to the differences in the types of outcomes in the included studies, a meta-analysis was not possible; therefore, we conducted a narrative synthesis of the data.

## Risk-of-bias and quality assessment

Two reviewers (NG, JL) independently assessed risk of bias for each included study using the Quality Assessment of Diagnostic Accuracy 2 (QUADAS-2) tool for diagnostic accuracy within each study [18], which was modified for self-assessment accuracy (see S2 File). The quality of each study was also rated using the Medical Education Research Study Quality Instrument (MERSQI) [19]. Two reviewers (MAS, CP) evaluated the following six domains of all included studies: study design; sampling; type of data; validity evidence for evaluation instrument scores; data analysis; and outcome. Using a weighting score for each section, an overall score that ranged between 5 and 18 was calculated.

## Results

Our initial search yielded 2,376 original articles, which included 279, 409, and 194 papers from subsequent updates, respectively. There were 1,302 duplicates that were removed. At the title and abstract screening stage, 1, 074 studies were evaluated. Additionally, the grey literature search did not identify any relevant records. The most common reason for the exclusion of records at the time of the abstract was due to lack of defined self-assessment accuracy. At the full-text screening stage, 22 studies were evaluated and 13 of these were excluded, with no disagreements between authors (for rationale of excluded full texts, see S3 File). Ultimately, 9 studies met inclusion criteria and were included in this systematic review. The search flow with details regarding full-text article exclusion is summarized in Fig 1.

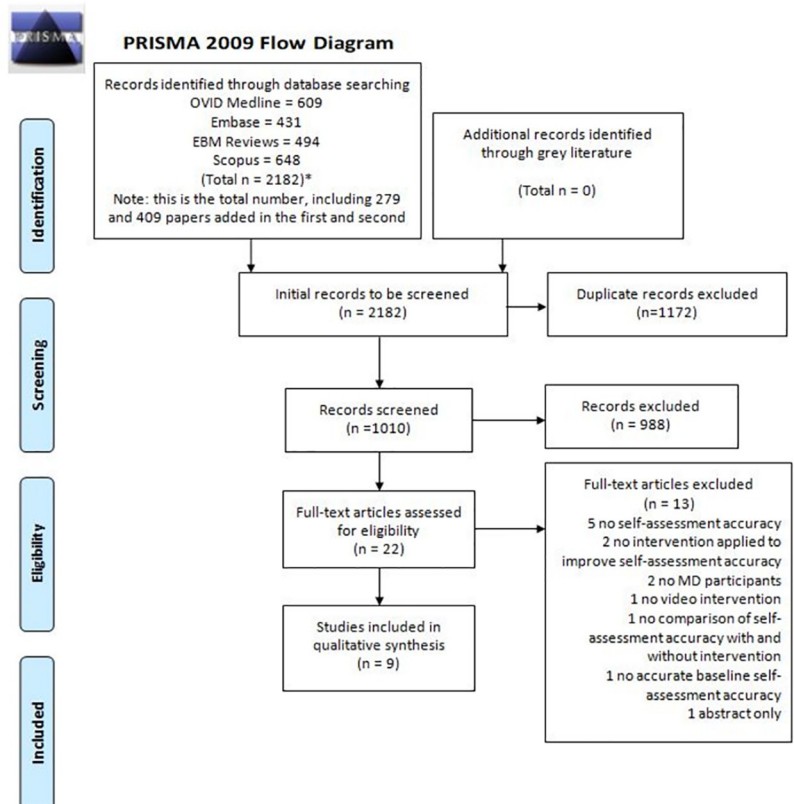

**Fig 1. Preferred Reporting Items for Systematic Reviews and Meta-Analyses flow chart illustrating the identification and selection process for studies included in this meta-analysis.**

## Study characteristics

The characteristics of included studies are outlined in Table 1. All 9 articles were interventional studies published between 1998 and 2019 [12,20–27], which included a total of 240 participants. All studies were single-center and 3 were randomized controlled trials while the remainder were non-randomized interventional studies. The most common specialty was General Surgery in 4/9 studies, one of which focused on Bariatric surgery while the remaining 3 represented core General Surgery. In terms of experience levels among the 240 participants, there were 17 (7.1%) medical students, 217 (90.4%) post-graduate trainees, and 6 (2.5%) attending physicians. All studies used a pre-post study design, with most using a single arm design (6/9). In terms of the type of skill, 5/9 studies investigated self-assessment of procedural skills including gastroscopy[7], sigmoidoscopy[12], laparoscopic surgery [21,25], and ultrasound-guided central venous catheter insertion into the internal jugular vein (US CVC IJ).[9] The remaining 4/9 studies were non-procedural, focused on tasks related to simulated patient encounters [24,28], resuscitations[10] and mock oral examinations [22].

## Characteristics of the interventions

Among the nine studies, the most common type of intervention was self-video review as evaluated in 8 studies, followed by benchmark video review in 4 studies and combination of benchmark and self-video review in 3 studies. Three of nine studies allowed for playback review with

Table 1. Characteristics for studies included in qualitative synthesis ($n = 9$).

| First author, year of publication | Speciality | Study design with time period between intervention and assessment | Procedure, overall skills assessed, and study setting | Total number of participants with level of training/experience | Number of participants in intervention arm(s) with summary of intervention | Number of participant in comparator arm(s) with summary of intervention | Type of self-assessment | Type of external assessment | Type of statistical analysis for primary outcome | Self-assessment accuracy at baseline, with directionality | Summary of intervention on self-assessment accuracy with P-value |
|---|---|---|---|---|---|---|---|---|---|---|---|
| Martin, 1998 [20] | Family Medicine | Assessment-intervention-assessment, single arm; no delay | Difficult patient scenario (e.g. child abuse), communication, simulated | 25 PGY1, 25 PGY-2 Family Medicine residents | 50 participants shown a videotape of four performances that ranged in quality from poor to good; Participants allowed to review the first 15 seconds of each performance | N/A | Evaluation form | Evaluation form completed by 2 communication experts and 4 academic family physicians | Cronbach's alpha and Pearson correlation | Moderately accurate (r = 0.38) | Benchmark video review: improved ($P<0.05$) |
| Ward, 2003 [21] | General Surgery | Assessment-intervention-assessment-intervention-assessment, single arm; no delay | Nissen fundoplication on live anesthetized pigs, technical skills, simulated | 25 PGY 3–5 residents and 1 fellow in General Surgery | 26 participants watched both videos of their own performance and then a benchmark video of that same performance in single session | N/A | Global rating scale and operative component rating scale | Global rating scale and operative component rating scale completed by 3 blinded faculty attending surgeons | Pearson Correlation | Moderately accurate (r = 0.50) | Benchmark video review: no change; Self-video review: improved ($P = 0.01$) |
| Kozol, 2004 [22] | General Surgery | Assessment-intervention-assessment, single arm; delay of up to one week | Mock oral board examinations, communication/professionalism, simulated | 7 PGY-3, 6 PGY-4, 7 PGY-5 General Surgery residents | 20 participants watched a video of their own performances | N/A | 5-point Likert scale | 5-point Likert scale completed by 2 faculty examiners | One way ANOVA of difference scores | Inaccurate, underestimation (Note: summative measure of accuracy not provided) | Self-video review: Improvement for specific parameters of organizational skills ($P = 0.007$), decision-making ($P = 0.033$), and professionalism ($P = 0.047$) |
| Sadosty, 2011 [23] | Emergency medicine | Assessment-intervention-assessment, single arm; 1 week delay | Management of two patients, clinical skills, simulated | 8 PGY-1,3 PGY-2, and 6 PGY-3 Emergency Medicine residents | 17 participants watched a video of their own performance immediately after the case | N/A | 16-item evaluation tool | 16-item evaluation tool completed by 3 unblinded Emergency Medicine staff | Logistic regression, linear regression, generalized estimation equation | Accurate (in 73.7% of items rated), high scorers more likely to accurately self-assess than low scorer ($P<0.001$) | Self-video review: No change ($P>0.05$) |
| Schellenberg,2014 [24] | Neurology | Assessment-intervention-assessment, single arm; no delay | Breaking bad news of amyotrophic lateral sclerosis, communication, simulated | 3 PGY-1,1 PGY-2, 6 PGY-3, 5 PGY-4, 7 PGY-5 Neurology residents | 22 participants watched a video of their own performance immediately after the case | N/A | Faculty developed checklist | Faculty developed checklist completed by 2 blinded Neurology staff | Generalized Estimation Equation for difference scores | Inaccurate, overestimation (GEE 0.145, p = 0.0012) | Self-video review: No change ($P>0.05$) |

(Continued)

**Table 1.** (Continued)

| First author, year of publication | Speciality | Study design with time period between intervention and assessment | Procedure, overall skills assessed, and study setting | Total number of participants with level of training/experience | Number of participants in intervention arm(s) with summary of intervention | Number of participant in comparator arm(s) with summary of intervention | Type of self-assessment | Type of external assessment | Type of statistical analysis for primary outcome | Self-assessment accuracy at baseline, with directionality | Summary of intervention on self-assessment accuracy with P-value |
|---|---|---|---|---|---|---|---|---|---|---|---|
| Herrera-Almario, 2015 [25] | General Surgery | Assessment-Intervention-assessment, single arm; 7 to 10 day delay | Laparoscopic surgical procedures, technical skills, clinical | 9 PGY-3 General Surgery Residents | 9 participants watched a video of their own performance 7 to 10 days after performing the procedure along with the supervising surgical attending without verbal feedback; Participants allowed to fast forward through segments of inactivity of surgical instruments. | N/A | GOALS tool [29] | GOALS tool completed by 1 unblinded Surgical staff | Difference score, paired t tests, Wilcoxon signed rank test | Inaccurate, underestimation ($p < 0.0001$ for all domains except suturing $p = 0.0471$) | Self-video review: No change ($P > 0.05$) |
| Wittler, 2016 [26] | Emergency Medicine | Assessment-intervention-assessment, two arm randomized trial; no delay | Ultrasound guided internal jugular central venous catheter placement, technical skills, simulated | 15 PGY-1 Emergency Medicine residents | An unspecified number of participants given standardized verbal feedback with video review of their own performance; Specific video segments were chosen by the expert to illustrate constructive points | An unspecified number of participants given standardized verbal feedback only | 30-point checklist with global rating scale | 30-point checklist with global rating scale completed by 2 unblinded faculty Emergency Medicine staff | Plotted difference scores, ordinary least squares linear regression | Inaccurate, higher scorers underestimated while lower scorers overestimated (Slope of regression line 0.60, p = 0.027) | Self-video review: No improvement compared to verbal feedback alone ($P > 0.05$) |
| Vyasa, 2017 [27] | General surgery | Assessment-Intervention-Practice-Assessment, three arm randomized trial; no delay | Sigmoidoscopy, technical skills, simulated | 30 PGY-1 General Surgery residents | 10 participants watched a video of their own performance, followed by simulator practice; 10 participants reviewed a benchmark video with note-taking, followed by simulator practice | 10 participants had only simulator practice | Computer metrics on simulator | Computer metrics on simulator | Difference scores, ANOVA, paired T-tests | Inaccurate, overestimation of time to complete task (59.8), arrive at cecum (47.1), time patient was in pain (14.6), and overall efficiency (20.1); underestimation of how much mucosal surface examined (-15.8) | Benchmark video review: improvement compared to self-video review and simulator practice only for overall procedural efficiency ($P < 0.01$) |

*(Continued)*

Table 1. (Continued)

| First author, year of publication | Speciality | Study design with time period between intervention and assessment | Procedure, overall skills assessed, and study setting | Total number of participants with level of training/experience | Number of participants in intervention arm(s) with summary of intervention | Number of participant in comparator arm(s) with summary of intervention | Type of self-assessment | Type of external assessment | Type of statistical analysis for primary outcome | Self-assessment accuracy at baseline, with directionality | Summary of intervention on self-assessment accuracy with P-value |
|---|---|---|---|---|---|---|---|---|---|---|---|
| Scaffidi, 2019 [12] | Gastroenterology | Assessment, intervention, assessment, assessment-assessment, three arm randomized trial; no delay | EGD, technical skills, simulated | 51 Novice endoscopists (<20 previous EGDs) with 17 medical students, 28 residents, and 6 staff | 17 participants watched a video of their own performance; 17 participants watched a video of a benchmark performance; Participants had 15 min to review videos and were allowed to rewind or fast forward as much as required in that time | 17 participants watched a video of both their own performance and of a benchmark performance | GIEGAT GRS [30] | GiECAT GRS completed by 2 blinded experts | Difference score, Bland-Altman analysis, Kruskal-Wallis, Friedman's test | Moderately accurate (ICC = 0.74) | Benchmark video review: Improvement compared to the benchmark and self-video review group ($P = 0.005$) Self-video review: no change ($P>0.05$) Benchmark and self-video review: improvement over time ($P = 0.016$) |

ANOVA, analysis of variance; EGD, esophagoduodenoscopy; GiECAT GRS, Gastrointestinal Endoscopy Competency Assessment Tool Global Rating Scale; GOALS, Global Operative Assessment of Laparoscopic Skills; ICC, intra-class correlation coefficient N/A, not applicable; PGY, post-graduate year; GEE, Generalized Estimating Equation.

fast-forwarding and/or rewinding throughout the video intervention. One study (1/9) allowed for experts to choose specific video segments for each participant to review in order to illustrate constructive feedback. Three (3/9) studies reported a delay between the task and video review. Overall, there was a paucity of information regarding the length of the video and characteristics of the video review in the majority of studies.

## Assessment of outcomes

The most common type of outcome assessment was an assessment tool with prior validity evidence, as used in 4 (4/9) studies [12,21,25,26]. External assessment was most commonly performed by experts who were unblinded to participant identity/group assignment (5/9); one (1/9) study used computer metrics on an endoscopy simulator. There was no time delay between the intervention and assessment of outcome in 6 (6/7) studies.

## Impact of video interventions on self-assessment accuracy

At baseline, 5 (5/9) studies found that there was inaccurate self-assessment among participants; three (3/5) of these studies focused on procedural skill. Of these 5 studies, 2 (General Surgery) reported under-estimation of skill, 1 (Neurology) reported over-estimation, and the remaining 2 reported indeterminate variations in self-assessment accuracy.

After the video-based intervention, 5 (5/9) studies found that there was a statistically significant improvement in self-assessment accuracy. In terms of intervention type, benchmark video review was effective in 3 (3/9) studies, a combination of self-video and benchmark video review was effective in 1 study (1/9) and self-video review was effective in 2 (2/9) studies. Among the studies in which the benchmark video review was effective, 2 were procedural (General Surgery, Gastroenterology) and 1 was non-procedural (General Surgery). In contrast, both (2/9) studies in which self-video review was effective were from General Surgery, only 1 of which was procedural.

There were 2 (2/9) studies that directly compared different types of video-based interventions using two different participant arms. In one study, benchmark video review led to a significant improvement in self-assessment accuracy when compared directly to the combination group of both benchmark video review and self video review; no significant differences were observed when benchmark video review was compared to self-video review [12]. In the other study, benchmark video review led to a significant improvement in self-assessment accuracy compared to self-video review and to simulator practice only [31].

Additionally, there was another study that compared the use of two video-based interventions within the same group over time, wherein self-assessment accuracy was measured after self-video review and then again after review of four benchmark performances [32]. This study found that self assessment accuracy only improved after self-video review and not benchmark video review.

## Risk of bias and study quality

Risk of bias assessments are summarized in Fig 2. Eight (8/9) studies had a low risk of bias and one (1/9) study had a high risk of bias. In the one study with high risk of bias, there was no formal randomization of participants noted and there was a potential for bias in how the reference standard used, as participants reviewed the video with the assessors, which could have introduced bias. MERSQI scores ranged from 11.5 to 13, suggesting a overall high study quality [33] (see Table 2).

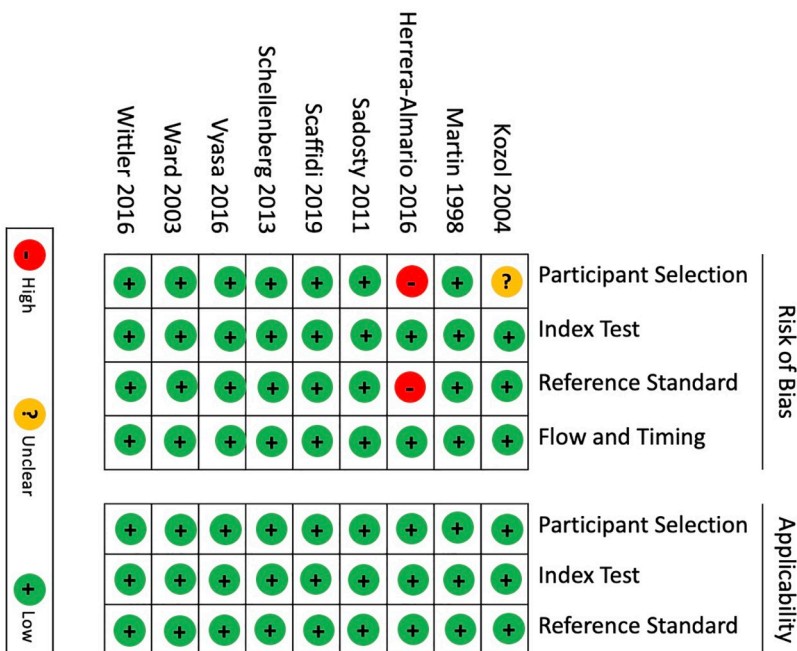

**Fig 2. Quality assessment of diagnostic accuracy studies 2 (QUADAS-2) risk of bias assessment.**

## Discussion

In this systematic review of video-based interventions to improve self-assessment accuracy of physicians, we found that most interventions were effective, as reported in 5 studies of the 9 included studies [12,20–27]. Video-based interventions used either videos of participants' own performances (i.e., self-video review) or, less commonly, expert performances (i.e., benchmark video review), or, least commonly of all, a combination of both. Overall, benchmark video review was the most effective intervention. The primary strength of our study is that it is the first systematic review on interventions of any kind aimed at improving self-assessment accuracy among physicians. Additionally, our search was comprehensive, as it included six databases and used broad search criteria.

One explanation for the finding that videos, especially benchmark video review, can be effective at improving self-assessment accuracy is that it addresses deficiencies outlined by the Dunning-Kruger effect. In particular, individuals who have less experience or training may not be able to develop an accurate schema of their performance. This discrepancy, however, can be mitigated by providing an external point of reference (e.g. viewing a benchmark performance). This explanation is reinforced by our finding that benchmark video review was the most effective type of intervention, as it significantly improved self-assessment accuracy in almost all of the studies that used it [12,20,27]. Furthermore, in two studies that directly compared with an intervention involving self-video review (or a combination of benchmark video review with self-video review) [27,34], benchmark video review was found to be more effective. Of note, however, one study that found that benchmark video review did not significantly improve self-assessment accuracy; in this same study, self-video review used prior to the benchmark did indeed lead to more accurate assessments [21].

The differences in the effectiveness of benchmark videos compared to self-review videos may be contingent on physician experience levels. In particular, the one study which found

**Table 2. Quality assessment for included studies (n = 9).**

| First author (ref.); year of publication | Study design (score [max 3]) | Sampling: Number of institutions (score [max 1.5]) | Sampling: follow-up (score [max 1.5]) | Type of data: Outcome assessment (score [max 3]) | Validity evidence for evaluation instrument scores (score [max 3]) | Data analysis: appropriate (score [max 1]) | Data analysis: sophistication (score [max 2]) | Highest outcome type (score [max 3]) | TOTAL MERSQI |
|---|---|---|---|---|---|---|---|---|---|
| Martin, 1998 [20] | Single-group pretest and posttest (1.5) | 1 institution (0.5) | ≥75% (1.5) | Objective (3) | Internal structure (1) | Data analysis appropriate for study design and type of data (1) | Beyond descriptive analysis (2) | Knowledge, skills (1.5) | 12 |
| Ward, 2003 [21] | Single-group pretest and posttest (1.5) | 1 institution (0.5) | ≥75% (1.5) | Objective (3) | Internal structure (1) | Data analysis appropriate for study design and type of data (1) | Beyond descriptive analysis (2) | Knowledge, skills (1.5) | 12 |
| Kozol, 2004 [22] | Single-group pretest and posttest (1.5) | 1 institution (0.5) | ≥75% (1.5) | Objective (3) | Content (1) | Data analysis appropriate for study design and type of data (1) | Beyond descriptive analysis (2) | Knowledge, skills (1.5) | 12 |
| Sadosty, 2011 [23] | Single-group pretest and posttest (1.5) | 1 institution (0.5) | ≥75% (1.5) | Objective (3) | Content (1) | Data analysis appropriate for study design and type of data (1) | Beyond descriptive analysis (2) | Knowledge, skills (1.5) | 12 |
| Schellenberg, 2014 [24] | Single-group pretest and posttest (1.5) | 1 institution (0.5) | 50% to 74% (1) | Objective (3) | Content (1) | Data analysis appropriate for study design and type of data (1) | Beyond descriptive analysis (2) | Knowledge, skills (1.5) | 11.5 |
| Herrera-Almario, 2015 [25] | Single-group pretest and posttest (1.5) | 1 institution (0.5) | ≥75% (1.5) | Objective (3) | Relationships to other variables (1) | Data analysis appropriate for study design and type of data (1) | Beyond descriptive analysis (2) | Knowledge, skills (1.5) | 12 |
| Wittler, 2016 [26] | Randomized controlled trial (3) | 1 institution (0.5) | ≥75% (1.5) | Objective (3) | Internal structure (1) | Data analysis appropriate for study design and type of data (1) | Beyond descriptive analysis (2) | Knowledge, skills (1.5) | 13 |
| Vyasa, 2017 [27] | Randomized controlled trial (3) | 1 institution (0.5) | ≥75% (1.5) | Objective (3) | Content (1) | Data analysis appropriate for study design and type of data (1) | Beyond descriptive analysis (2) | Knowledge, skills (1.5) | 13 |
| Scaffidi, 2019 [12] | Randomized controlled trial (3) | 1 institution (0.5) | ≥75% (1.5) | Objective (3) | Internal structure (1) | Data analysis appropriate for study design and type of data (1) | Beyond descriptive analysis (2) | Knowledge, skills (1.5) | 13 |

*MERSQI*, Medical Education Research Study Quality Instrument.

self-video review more effective than benchmark-video review, was conducted among senior surgical residents [21]. The authors posited that this finding was likely due to the experience level of the residents–that is, a benchmark video did not provide them with any new information [21]. Rather, the residents likely already had an accurate working schema of their abilities that was only improved when they had an objective record of their own performances (i.e.,

self-video review) to inform their self-assessments. To this end, the effectiveness of benchmark videos is likely influenced by prior experience, with more novice physicians benefiting most. This is reflective of the process of "modelling", which is well described in the psychological literature [35]. More specific to medical education, the use observation of videos to learn clinical skills has been demonstrated in several studies. For example, video-recorded performances, even when containing errors, supply learners with an idea of the variability of a performance's scope [36].

We note the study limitations. Primarily, we could not quantify the impact of videos on self-assessment with a meta-analysis due to the lack of standardized outcome measures and different types of tasks in the included studies. In particular, the Bland-Altman analysis, a well described approach in the method comparison literature [37], is preferred, as this can allow for meta-analytic summation [8,38]. Similarly, we could not account for the potential confounding effect of accurate self-assessment at baseline, as noted in four studies. In addition, most studies did not quantify intervention length, which restricted our ability to determine whether length of the video interventions influences outcomes. The other characteristic we could not control for was physician level of experience and specialty that preceded the intervention, especially as the former has shown to have a significant impact on physician self-assessment [8]. Third, we may have missed relevant studies due to language bias, as we only included English-language abstracts, and due to a lack of universal definition of self-assessment accuracy. Last, we cannot definitively determine whether publication bias was present, as there may be additional studies with negative findings that were not yet published.

## Conclusions

In closing, our systematic review highlights several implications for medical education. First, benchmark video review may enable physicians to improve self-assessment accuracy, especially for those with limited experience in performing a particular clinical skill. Similarly, self-video review may be able to inform more experienced physicians' self-assessments. Second, due to the absence of similar analyses that precluded a meta-analysis, we suggest that future studies use standardized methods of comparison for self-assessment accuracy. Finally, it would be helpful for future research to delineate the effect of video-based interventions on both self-assessment accuracy and skill acquisition–that is, whether there is a relationship between improving self-assessment accuracy on skill acquisition. Quantifying this relationship would advance the objective impact of accurate self-assessment on clinical practice.

## Supporting information

**S1 Checklist. PRISMA 2020 checklist.**
(PDF)

**S1 File. Search strategy.**
(DOCX)

**S2 File. QUADAS-2 tool modified for self-assessment accuracy studies.**
(DOCX)

**S3 File. Reference list for excluded studies from full-text review, with respective rationale for exclusion.**
(DOCX)

## Author Contributions

**Conceptualization:** Chandni Pattni, Michael Scaffidi, Nikko Gimpaya, Catharine M. Walsh, Samir C. Grover.

**Data curation:** Chandni Pattni, Michael Scaffidi, Juana Li, Shai Genis, Nikko Gimpaya, Nazi Torabi.

**Formal analysis:** Chandni Pattni, Michael Scaffidi, Juana Li, Shai Genis, Nikko Gimpaya.

**Methodology:** Chandni Pattni, Michael Scaffidi, Samir C. Grover.

**Project administration:** Samir C. Grover.

**Supervision:** Catharine M. Walsh, Samir C. Grover.

**Writing – original draft:** Chandni Pattni, Michael Scaffidi, Rishi Bansal, Nazi Torabi, Catharine M. Walsh, Samir C. Grover.

**Writing – review & editing:** Chandni Pattni, Michael Scaffidi, Rishad Khan, Catharine M. Walsh, Samir C. Grover.

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
