## [Decision Letter · Decision Letter 0]

2 Jan 2023

PONE-D-22-30231Video-based interventions to improve self-assessment accuracy among physicians: A systematic reviewPLOS ONE

Dear Dr. Pattni,

Thank you for submitting your manuscript to PLOS ONE. After careful consideration, we feel that it has merit but does not fully meet PLOS ONE’s publication criteria as it currently stands. Therefore, we invite you to submit a revised version of the manuscript that addresses the points raised during the review process.

We look forward to receiving your revised manuscript.

Kind regards,

Florian Recker, M.D., MME

Academic Editor

PLOS ONE

Journal Requirements:

   "Other disclosures: Rishad Khan has received research grants from AbbVie and Ferring Pharmaceuticals and research funding from Pendopharm. Samir C. Grover has received research grants and personal fees from AbbVie and Ferring Pharmaceuticals, personal fees from Takeda, education grants from Janssen, and has equity in Volo Healthcare."

3. Please include your tables as part of your main manuscript and remove the individual files. Please note that supplementary tables (should remain/ be uploaded) as separate "supporting information" files

Additional Editor Comments:

The language needs some revision as sentences are often very long and therefore difficult to read.

A more detailed analysis of the included and excluded criteria muste be mentioned. For more detailed information the amount of participants in the nine included studies should be added to the study characteristics in details.

It should be considered to display and discuss the results of the different study types, e.g. studies for enhancing practical skills and studies for improving verbal/communication skills separately.

The percentage of over- vs. under-estimation of performance in the different studies and study-types would be interesting.

The identified risks of bias could be explained more clearly in the text.

Reviewers' comments:

Reviewer's Responses to Questions

**Comments to the Author**

1. Is the manuscript technically sound, and do the data support the conclusions?

Reviewer #1: Yes

Reviewer #2: Partly

2. Has the statistical analysis been performed appropriately and rigorously?

Reviewer #1: N/A

Reviewer #2: Yes

3. Have the authors made all data underlying the findings in their manuscript fully available?

Reviewer #1: Yes

Reviewer #2: Yes

4. Is the manuscript presented in an intelligible fashion and written in standard English?

Reviewer #1: Yes

Reviewer #2: No

5. Review Comments to the Author

Reviewer #1: This review assesses the current evidence about the efficacy of video-based interventions to improve self-assessment accuracy on medical education.

The manuscript is very well written. Research methodology and selection process can both be clearly understood. Notwithstanding this, this reviewer has to express his doubt about exhaustivity of study selection. The authors state: „We only examined studies of physicians (and not medical students), as we aimed …“. Although the Strudy of Scaffidi et al includes 35-47% of students in each arm. How do the authors support this inclusion?

On my opinion the low amount of studies included, assessing very different, heterogeneous medical skills among physicians with different experience levels does not allow to support a general statement about the trained technique as the authors try to support. Since the method evaluation of the different studies does not allow comparison, it seems uncareful to asses a statement about the overall performance of this intervention. The authors describe briefly some effect of experience of the physician in the discussion. Although, this reviewer would suggest putting more effort explaining effect size of each study or the relevance of experience.

Because of all this, on my opinin this paper does not meet the standards to be considered for publication.

Reviewer #2: The language needs some revision as sentences are often very long and therefore difficult to read.

Exclusion criteria of the 988 excluded studies should be mentioned in more detail. Why were so many studies excluded? How many studies were excluded because of the lack of what kind of data?

The characteristics of the Interventions (line 236-238) do not add up to 100%. For more detailed information the amount of participants in the nine included studies should be added to the study characteristics in details.

The included studies should be characterised in more detail. It would be more informative for the reader to mention the surgical disciplines as well as the number of participants and their expertise not only in the table but also as a sum-up in the text. It might be interesting to display if there is an under-or overestimation of skills and what disciplines tend towards which direction.

It should be considered to display and discuss the results of the different study types, e.g. studies for enhancing practical skills and studies for improving verbal/communication skills separately.

The percentage of over- vs. under-estimation of performance in the different studies and study-types would be interesting.

The identified risks of bias could be explained more clearly in the text.

6. PLOS authors have the option to publish the peer review history of their article (what does this mean?). If published, this will include your full peer review and any attached files.

**Do you want your identity to be public for this peer review?** For information about this choice, including consent withdrawal, please see our Privacy Policy.

Reviewer #1: No

Reviewer #2: No

---

## [Author Response · Author response to Decision Letter 0]

18 Feb 2023

February 15, 2023, 

Dr. Emily Chenette

Editor-in-chief, PLOS ONE

Dear Dr. Chenette and members of the editorial board:

Thank you for giving us the opportunity to revise and resubmit our manuscript (PONE-D-22-30231) entitled “Video-based interventions to improve self-assessment accuracy among physicians: A systematic review.” We thank the reviewers for their insightful and thoughtful comments. We have addressed all of the reviewers’ queries and have outlined our responses and any corresponding changes below: 

EDITORIAL COMMENTS

We have addressed all style requirements, including for file naming. 

(2) Thank you for stating the following in the Competing Interests section: 

 "Other disclosures: Rishad Khan has received research grants from AbbVie and Ferring Pharmaceuticals and research funding from Pendopharm. Samir C. Grover has received research grants and personal fees from AbbVie and Ferring Pharmaceuticals, personal fees from Takeda, education grants from Janssen, and has equity in Volo Healthcare." Please confirm that this does not alter your adherence to all PLOS ONE policies on sharing data and materials, by including the following statement: "This does not alter our adherence to PLOS ONE policies on sharing data and materials.” Please include your updated Competing Interests statement in your cover letter; we will change the online submission form on your behalf.

We have added the required statement to the Cover Letter.

(3) Please include your tables as part of your main manuscript and remove the individual files. Please note that supplementary tables (should remain/ be uploaded) as separate "supporting information" files

We have included the tables in the body of the manuscript and as separate files as requested.

(4) Please include captions for your Supporting Information files at the end of your manuscript, and update any in-text citations to match accordingly. 

We have added the appropriate captions. 

REVIEWER #1

(1)The manuscript is very well written. Research methodology and selection process can both be clearly understood. Notwithstanding this, this reviewer has to express his doubt about exhaustivity of study selection. The authors state: „We only examined studies of physicians (and not medical students), as we aimed …“. Although the Strudy of Scaffidi et al includes 35-47% of students in each arm. How do the authors support this inclusion?

We agree that this point requires clarification. Our intention in seeking only to examine studies of physicians was that we wanted to represent a narrower group, as we do not consider medical students to be representative of fully licensed physicians. Nevertheless, the inclusion of the study by Scaffidi et al that has a proportionally significant amount of medical students does call this criterion into question. To clarify this, we have added a qualifier in the Methods that we sought to include studies in which the majority of participants were physicians (pg 7). 

(2) On my opinion the low amount of studies included, assessing very different, heterogeneous medical skills among physicians with different experience levels does not allow to support a general statement about the trained technique as the authors try to support. Since the method evaluation of the different studies does not allow comparison, it seems uncareful to asses a statement about the overall performance of this intervention. The authors describe briefly some effect of experience of the physician in the discussion. Although, this reviewer would suggest putting more effort explaining effect size of each study or the relevance of experience.

We also agree that there are heterogeneous medical skills among physicians with different experience levels. We have commented on this as a limitation in the Discussion (pg. 14). Additionally, it is true that there are different measures of effects (which we have highlighted already as one of the limitations). 

REVIEWER #2

(1) The language needs some revision as the sentences are often very long and therefore difficult to read.

We have addressed this point by reviewing the manuscript and making changes as appropriate.

(2) Exclusion criteria of the 988 excluded studies should be mentioned in more detail. Why are so many studies excluded? How many studies were excluded because of the lack of what kind of data?

We agree that many studies were excluded. Most of these studies, however, were excluded because they failed to properly evaluate self-assessment accuracy with comparison of self-assessment scores to an external observer. We have acknowledged this point in the Results section (pg. 9).

(3) The characteristics of the interventions (line 236-238) do not add up to 100%.

We acknowledge that there was an error in the calculation, which is corrected in the Results section (pg. 10). Nevertheless, we still would not expect that the numbers add up to 100% as there is overlap between the interventions. For example, self-video was evaluated in 8 of 9 studies but 3 of the 8 studies also involved benchmark video intervention. The combination of benchmark and self-video interventions was evaluated 3 of 9 studies, while benchmark was evaluated in 4 of 9 studies. Therefore, only 1 study exclusively evaluated the benchmark video intervention. The purpose of this line was to highlight that self-video was the most common video intervention. 

(4) More detailed information about the nine included studies should be added to the study characteristics in details. It would be more informative for the reader to mention the surgical disciplines as well as the number of participants and their expertise not only in the table but also as a sum-up in the text.

We agree that these details would be beneficial and have added them in the Results (pg. 10). 

(5) It might be interesting to display if there is an under-or overestimation of skills and what disciplines tend towards which direction. The percentage of over- vs. under-estimation of performance in the different studies and study-types would be interesting.

We agree that these details would be useful to demonstrate and have incorporated the recommendations into the Results (pg 11). 

(6) It should be considered to display and discuss the results of the different study types, e.g. studies for enhancing practical skills and studies for improving verbal/communication skills separately.

We have provided the requested context in the Results section (pg. 11). 

(7) The identified risks of bias could be explained more clearly in the text. (MS)

We have explained the risk of bias more clearly in the Results section (pg. 12). 

Thank you once again for your consideration of this manuscript. We look forward to your review and comments.

Yours sincerely,

Samir C. Grover, MD, MEd, FRCPC

Division of Gastroenterology

St. Michael’s Hospital

---

## [Decision Letter · Decision Letter 1]

8 May 2023

PONE-D-22-30231R1Video-based interventions to improve self-assessment accuracy among physicians: A systematic reviewPLOS ONE

Dear Dr. Pattni,

Thank you for submitting your manuscript to PLOS ONE. After careful consideration, we feel that it has merit but does not fully meet PLOS ONE’s publication criteria as it currently stands. Therefore, we invite you to submit a revised version of the manuscript that addresses the points raised during the review process.

ACADEMIC EDITOR: in case of small samples, percentages are not recommended to be reported; use instead the ration (e.g. 3/9)self assessment of what? cooking? physical examination? please be specific.please avoid in scientific writing the following expressions "There is/are".the aim: any kind of physicians?"limited among physicians and surgeons" please define limited.lacking in which way?An update of PRISMA, namely PRISMA-2020, exists and should be followed."PHYSICIANS" used in the search string would not retrieve manuscripts with "physician"The main reason for exclusion at screening is lack of definition of accuracy; this is the main issue of your study because it is well known that such detailed information could not necessary stay in the abstract of the manuscript.Please list the other reasons for exclusion.The reason of exclusions must stay in the prism flowchart not in supplementary file."was an assessment tool with prior validity evidence, as used in 4 (44.4%) studies." please provide the references for these studies.The exclusion of studies in which no clear definition of accuracy was present in the abstract is a major limitation. Furthermore, inclusion of studies written in English is another limitation.You include studies with pre and post assessment; it is unclear why this information is reported in Table 1.Please do not duplicate the results in text and tables.In Table 2 it is not clear how the scores were attributed. A solution is to present briefly the instrument in the Material and Methods section.Please submit your revised manuscript by Jun 22 2023 11:59PM. If you will need more time than this to complete your revisions, please reply to this message or contact the journal office at plosone@plos.org. Please include the following items when submitting your revised manuscript:A rebuttal letter that responds to each point raised by the academic editor and reviewer(s). You should upload this letter as a separate file labeled 'Response to Reviewers'.A marked-up copy of your manuscript that highlights changes made to the original version. You should upload this as a separate file labeled 'Revised Manuscript with Track Changes'.An unmarked version of your revised paper without tracked changes. You should upload this as a separate file labeled 'Manuscript'.If applicable, we recommend that you deposit your laboratory protocols in protocols.io to enhance the reproducibility of your results. Protocols.io assigns your protocol its own identifier (DOI) so that it can be cited independently in the future. For instructions see: https://journals.plos.org/plosone/s/submission-guidelines#loc-laboratory-protocols. Additionally, PLOS ONE offers an option for publishing peer-reviewed Lab Protocol articles, which describe protocols hosted on protocols.io. Read more information on sharing protocols at https://plos.org/protocols?utm_medium=editorial-email&utm_source=authorletters&utm_campaign=protocols.

We look forward to receiving your revised manuscript.

Kind regards,

Sorana D. Bolboacă, Ph.D., M.Sc., M.D.

Academic Editor

PLOS ONE

Journal Requirements:

Reviewers' comments:

Reviewer's Responses to Questions

**Comments to the Author**

1. If the authors have adequately addressed your comments raised in a previous round of review and you feel that this manuscript is now acceptable for publication, you may indicate that here to bypass the “Comments to the Author” section, enter your conflict of interest statement in the “Confidential to Editor” section, and submit your "Accept" recommendation.

Reviewer #2: (No Response)

2. Is the manuscript technically sound, and do the data support the conclusions?

Reviewer #2: Yes

3. Has the statistical analysis been performed appropriately and rigorously? 

Reviewer #2: Yes

4. Have the authors made all data underlying the findings in their manuscript fully available?

Reviewer #2: Yes

5. Is the manuscript presented in an intelligible fashion and written in standard English?

Reviewer #2: No

6. Review Comments to the Author

Reviewer #2: The language still needs some revision as some words are used incorrectly and sentences do not make sense. For example:

347 Quantifying this relationship would advance the objective impact of accurate self-assessment on

clinical practice.

63 Self-assessment can be used to identify physicians’ deficiencies to allow for appropriate corrective measures.

102 Not only can inaccurate self-assessment impair opportunities for growth, but overconfidence stemming from one’s own performances can lead to patient safety concerns.

Disagreements were resolved by consensus.

256 In contrast, of the 2 studies in which self-video review was effective, both were from General

Surgery and 1 was procedural.

The Conclusions in the Abstract should describe the results of the manuscript in more detail, for example:

Benchmark video review may enable physicians to improve self-assessment accuracy, especially for those with limited experience in performing a particular clinical skill. Self-video review may be able to improve the self-assessment of more experienced physicians’.

Citing of Literature should be avoided in the Conclusions (omit the sentence in line 342-344 and put it in the right context in the discussion).

The first sentence of the Conclusions should be omitted as it does not provide a conclusion. Suggestion: Our review highlights the following suggestions for medical education:...

7. PLOS authors have the option to publish the peer review history of their article (what does this mean?). If published, this will include your full peer review and any attached files.

Reviewer #2: No

---

## [Author Response · Author response to Decision Letter 1]

29 May 2023

May 29, 2023 

Dr. Emily Chenette

Editor-in-chief, PLOS ONE

Dear Dr. Chenette and members of the editorial board:

Thank you for giving us the opportunity to revise and resubmit our manuscript (PONE-D-22-30231) entitled “Video-based interventions to improve self-assessment accuracy among physicians: A systematic review.” We thank the reviewers for their insightful and thoughtful comments. We have addressed all the reviewers’ queries and have outlined our responses and any corresponding changes below: 

ACADEMIC EDITOR COMMENTS

(1) In case of small samples, percentages are not recommended to be reported; use instead the ration (e.g. 3/9)

We agree with this recommendation and have removed the percentages throughout the manuscript. 

(2) Self assessment of what? cooking? physical examination? please be specific.

We have addressed this comment in the manuscript by specifying in the abstract that we are investigating self-assessment of a physician’s own performance on both procedural and non-procedural tasks in a given speciality of medicine or surgery.

(3) Please avoid in scientific writing the following expressions "There is/are".

We have removed the expression in the revised manuscript.

(4) The aim: any kind of physicians?

Yes, we were intentionally inclusive for several reasons. First, there is a general paucity of evidence in the first place for interventions aimed at improving self-assessment accuracy. Second, although there is heterogeneity among all physicians based on specialty, practice area and demographics, we conceptualize the practice of self-assessment on procedural tasks as a rather homogeneous activity, owing to its constrained parameters, i.e. assessment of one’s own performance of an external task.

(5) "limited among physicians and surgeons" please define limited.

We have clarified this with the term “inaccurate” In the Introduction (pg 4).

(6) lacking in which way?

We have clarified that there had not been a systematic evaluation of the literature until our present manuscript. 

(7) An update of PRISMA, namely PRISMA-2020, exists and should be followed.

An updated and completed PRISMA-2020 checklist had been uploaded with the original manuscript submission. We have provided the updated reference.

(8) "PHYSICIANS" used in the search string would not retrieve manuscripts with "physician"

Our search strategy was created by a qualified health research librarian employed by the University of Toronto Gerstein Library. She has assured us that there would not be any manuscripts missing with the use of “PHYSICIANS” as opposed to “physician.”

(9) The main reason for exclusion at screening is lack of definition of accuracy; this is the main issue of your study because it is well known that such detailed information could not necessary stay in the abstract of the manuscript.

We kindly refer the Editor to the Definitions section of the Methods (pg 5), wherein we explicitly define what we mean by self-assessment accuracy, which is “direct comparison between an external evaluator and self-assessment that was quantified using formal statistical analysis”. We have clarified this in the Abstract (pg 2).

(10) Please list the other reasons for exclusion.

We have clarified the other reasons for exclusions in the Methods (pg 6).

(11) The reason of exclusions must stay in the prisma flowchart not in supplementary file.

We have not provided the list of reasons for exclusions, but rather put the full text references excluded.

(12) "was an assessment tool with prior validity evidence, as used in 4 (44.4%) studies." please provide the references for these studies.

We have added the relevant references in the Results (pg 10).

(13) The exclusion of studies in which no clear definition of accuracy was present in the abstract is a major limitation. Furthermore, inclusion of studies written in English is another limitation.

We have added these two limitations in the Discussion (pp 13-14).

(14) You include studies with pre and post assessment; it is unclear why this information is reported in Table 1.

We have removed redundant information in Table 1. 

(15) Please do not duplicate the results in text and tables.

Initially, our draft manuscript did not duplicate this; we, however, added in the data in Table form at the request of the reviewers.

(16) In Table 2 it is not clear how the scores were attributed. A solution is to present briefly the instrument in the Material and Methods section.

We have added in a brief description of the MERSQI tool in the Material and Methods (pg. 8).

REVIEWER #2

(1) The language still needs some revision as some words are used incorrectly and sentences do not make sense. For example:

347 Quantifying this relationship would advance the objective impact of accurate self-assessment on

clinical practice.

63 Self-assessment can be used to identify physicians’ deficiencies to allow for appropriate corrective measures.

102 Not only can inaccurate self-assessment impair opportunities for growth, but overconfidence stemming from one’s own performances can lead to patient safety concerns.

Disagreements were resolved by consensus.

256 In contrast, of the 2 studies in which self-video review was effective, both were from General Surgery and 1 was procedural.

We have addressed these points by reviewing the manuscript and making changes as appropriate.

(2) The Conclusions in the Abstract should describe the results of the manuscript in more detail, for example:

Benchmark video review may enable physicians to improve self-assessment accuracy, especially for those with limited experience in performing a particular clinical skill. Self-video review may be able to improve the self-assessment of more experienced physicians’.

We appreciate the suggestion and have incorporated this into our abstract conclusion.

(3) Citing of Literature should be avoided in the Conclusions (omit the sentence in line 342-344 and put it in the right context in the discussion).

The first sentence of the Conclusions should be omitted as it does not provide a conclusion. Suggestion: Our review highlights the following suggestions for medical education.

We have addressed this point by reviewing the manuscript and making changes as appropriate.

Thank you once again for your consideration of this manuscript. We look forward to your review and comments.

Yours sincerely,

Samir C. Grover, MD, MEd, FRCPC

Division of Gastroenterology

St. Michael’s Hospital

---

## [Decision Letter · Decision Letter 2]

28 Jun 2023

Video-based interventions to improve self-assessment accuracy among physicians: A systematic review

PONE-D-22-30231R2

Dear Dr. Pattni,

We’re pleased to inform you that your manuscript has been judged scientifically suitable for publication and will be formally accepted for publication once it meets all outstanding technical requirements.

Kind regards,

Sorana D. Bolboacă, Ph.D., M.Sc., M.D.

Academic Editor

PLOS ONE

Additional Editor Comments:

Agree to disagree: "PHYSICIANS" vs. "PHYSICIAN"

Reviewers' comments:

Reviewer's Responses to Questions

**Comments to the Author**

1. If the authors have adequately addressed your comments raised in a previous round of review and you feel that this manuscript is now acceptable for publication, you may indicate that here to bypass the “Comments to the Author” section, enter your conflict of interest statement in the “Confidential to Editor” section, and submit your "Accept" recommendation.

Reviewer #2: All comments have been addressed

2. Is the manuscript technically sound, and do the data support the conclusions?

Reviewer #2: Yes

3. Has the statistical analysis been performed appropriately and rigorously? 

Reviewer #2: Yes

4. Have the authors made all data underlying the findings in their manuscript fully available?

Reviewer #2: Yes

5. Is the manuscript presented in an intelligible fashion and written in standard English?

Reviewer #2: Yes

6. Review Comments to the Author

Reviewer #2: My comments have been addressed. No further changes are needed. I recommend this paper for publication.

7. PLOS authors have the option to publish the peer review history of their article (what does this mean?). If published, this will include your full peer review and any attached files.

Reviewer #2: No

---

## [Editor Report · Acceptance letter]

4 Jul 2023

PONE-D-22-30231R2 

Video-based interventions to improve self-assessment accuracy among physicians: A systematic review 

Dear Dr. Pattni:

I'm pleased to inform you that your manuscript has been deemed suitable for publication in PLOS ONE. Congratulations! Your manuscript is now with our production department. 

Kind regards, 

on behalf of

Professor Sorana D. Bolboacă 

Academic Editor

PLOS ONE